# Differential Expression of Decorin in Metastasising Colorectal Carcinoma Is Regulated by *miR-200c* and Long Non-Coding RNAs

**DOI:** 10.3390/biomedicines10010142

**Published:** 2022-01-10

**Authors:** Margareta Žlajpah, Kristian Urh, Jan Grosek, Nina Zidar, Emanuela Boštjančič

**Affiliations:** 1Institute of Pathology, Faculty of Medicine, University of Ljubljana, 1000 Ljubljana, Slovenia; margareta.zlajpah@mf.uni-lj.si (M.Ž.); kristian.urh@mf.uni-lj.si (K.U.); nina.zidar@mf.uni-lj.si (N.Z.); 2Department of Abdominal Surgery, University Medical Centre Ljubljana, 1000 Ljubljana, Slovenia; jan.grosek@kclj.si; 3Faculty of Medicine, University of Ljubljana, 1000 Ljubljana, Slovenia

**Keywords:** colorectal carcinoma, liver metastasis, lymph node metastasis, decorin, *miR-200c*, lncRNA

## Abstract

Decorin (DCN) is one of the matricellular proteins that participate in normal cells’ function as well as in cancerogenesis. While its expression in primary tumours is well known, there is limited data about its expression in metastases. Furthermore, the post-transcriptional regulation of *DCN* is still questionable, although it is well accepted that it is an important mechanism of developing metastatic cancer. The aim of our study was to analyse the expression of DCN and its potential regulatory ncRNAs in metastatic colorectal carcinoma (CRC). Nineteen patients with metastatic CRC were included. Using qPCR, we analysed the expression of *DCN*, *miR-200c* and five lncRNAs (*LUCAT1*, *MALAT1*, *lncTCF7*, *XIST*, and *ZFAS1*) in lymph node and liver metastases in comparison to the invasive front and central part of a primary tumour. Our results showed insignificant upregulation of *DCN* and significant upregulation for *miR-200c*, *MALAT1*, *lncTCF7* and *ZFAS1* in metastases compared to the primary tumour. *miR-200c* showed a positive correlation with *DCN*, and the aforementioned lncRNAs exhibited a significant positive correlation with *miR-200c* expression in metastatic CRC. Our results suggest that *DCN* as well as *miR-200c*, *MALAT1*, *lncTCF7* and *ZFAS1* contribute to the development of metastases in CRC and that regulation of *DCN* expression in CRC by ncRNAs is accomplished in an indirect manner.

## 1. Introduction

Decorin (DCN) is one of the matricellular proteins belonging to the small leucine-rich proteoglycan family. DCN is transcribed and translated by fibroblasts, stressed vascular endothelial cells and smooth muscle cells [1]. It was first discovered as a protein-decorating collagen fibre [2], but today it is known as an important regulator of collagen fibrillogenesis [3,4]. Moreover, DCN participates in other cellular functions such as migration, proliferation, apoptosis and differentiation [5]. DCN also binds to growth factors, such as TGFβ, but on the other side, it inhibits receptor tyrosine kinases, e.g., EGFR, IGF-IR and met [1,6]. Consequently, the latter bioactivities have been attributed to evoke potent tumour repression [1].

DCN is one of many proteins in the extracellular matrix that provides not only physical scaffolds but also regulates many cellular processes, including cancerogenesis [7]. Several studies reported that a lack of DCN leads to spontaneous tumour development [2]. It has been shown that the expression of DCN is reduced in many carcinomas, e.g., in the ovaries, lung, oesophagus and others [2,4,8,9], where DCN is mainly synthesised by stromal cells, whereas it is almost completely absent in tumorous cells [9] or rarely by actively dividing normal cells [3]. The expression of DCN was also shown in some studies to be reduced in colorectal carcinoma (CRC) [3,4]; however, we [10] and others [9] reported a higher expression of DCN in CRC. We also showed that the *DCN* gene is upregulated regardless of the presence of lymph node metastasis, and it was the only gene significantly associated with the presence of lymph node metastasis in CRC. In the same study, we also observed an altered expression of *DCN* at the invasive front of CRC compared to its central part [10]. The extracellular matrix, including DCN, is used by disseminating tumorous cells to insulate themselves from the insult of haemodynamic fluid shear stress in circulation and to contribute to the establishment of a premetastatic niche by reorganizing the pre-existing extracellular matrix structure [11].

Post-transcriptional regulation of DCN is still questionable, although it is well accepted that its aberrant expression is important in developing lymph node and liver metastases in cancer, including CRC [12]. There are numerous known mechanisms of post-transcriptional regulation of mRNAs, the most frequently analysed being non-coding RNA (ncRNA) [13,14,15,16]. Among known groups of ncRNAs, miRNAs (microRNAs) and lncRNAs (long non-coding RNAs) have proven regulatory function and contribution to the development of metastasis in different carcinomas [17,18]. Up until now, no miRNA was confirmed to regulate *DCN*, and one miRNA, i.e., *miR-200c*, was predicted using bioinformatics analysis [19]. However, in vitro or in vivo correlation between expression of *miR-200c* and *DCN* has not been shown yet.

None of the lncRNAs have yet been proven to directly regulate *DCN*. In contrast, there are numerous proven lncRNAs that are validated to sponge *miR-200c* in different cancers, the predicted regulatory miRNA of *DCN* [20,21,22,23,24,25,26,27,28]. However, their exact role in regulating *miR-200c* in CRC has yet to be elucidated. Furthermore, while there are numerous studies describing the expression of *DCN* in CRC [3,4,5,9,12], there is limited data about the expression of mRNA *DCN* in lymph node and liver metastases. So far, only one study [29] reported a lower expression of the protein DCN in liver metastases of CRC.

Therefore, the aim of our study was to analyse the expression of *DCN* in the central part and the invasive front in comparison to the lymph node and liver metastases to further explore its role in advanced CRC. Furthermore, we tried to indirectly confirm previously predicted regulation of *DCN* on mRNA level by *miR-200c* [19] and to analyse whether experimentally validated lncRNAs might act as sponges for *miR-200c*.

## 2. Materials and Methods

Tissue samples from patients with CRC with lymph node and/or liver metastases were included in the study. For routine histopathologic examination, all tissue samples were fixed for 24 h in 10% buffered formalin and embedded in paraffin (FFPE). After fixation and embedding, tissues were cut into 4 µm slides and stained with haematoxylin and eosin for routine histopathological examination and classification according to pTNM (pathologic Tumour Node Metastasis). Samples were collected retrospectively from the archives of the Institute of Pathology, Faculty of Medicine, University of Ljubljana. On the basis of clinical and histopathological features, samples were divided into three groups: patients with CRC with lymph node metastases, patients with liver metastases but without lymph node metastases and patients with carcinoma with lymph node and liver metastases.

The investigation was carried out following the rules of the Declaration of Helsinki. The study was approved by the National Medical Ethics Committee (Republic of Slovenia, Ministry of Health).

### 2.1. Isolation of Total RNA from Tissue Cores

Tissue cores from the nodal and liver metastasis of CRC were punched from FFPE tissue blocks using a 600 µm needle. MagMax FFPE DNA/RNA Ultra kit (Applied Biosystems, Thermo Fisher Scientific, Inc.; Waltham, MA, USA) was used to extract the total RNA from three punches for each isolation. The total RNA extraction was isolated following the manufacturer’s instructions, with one modification: protease digestion was performed overnight with mixing for 15 s at 300 rpm every 4 min instead of 1 h. Apart from the deparaffinization solution (Xylene; Sigma-Aldrich; St. Louis, MO, USA) and the ethanol (Merck KGaA; Darmstadt, Germany), all the reagents were from Applied Biosystems (Thermo Fisher Scientific, Inc.). The quantity of RNA was measured with NanoDrop ND-1000 (Thermo Fisher Scientific, Inc.) by measuring the absorbance at 260 nm.

### 2.2. Selection of lncRNAs

After searching through the publication, we have identified 9 lncRNAs that were suggested to sponge *miR-200c* in different types of cancer or in other diseases [20,21,22,23,24,25,26,27,30]. Among the identified lncRNAs, we have chosen those that could successfully detect specific products of less than 100 bp using TaqMan probe. For the purpose of expression analysis, we, therefore, selected the following lncRNAs: *LUCAT1*, *MALAT1*, *lncTCF7*, *XIST* and *ZFAS1* [20,21,22,27,28]. The other four lncRNAs, namely *ATB*, *LINC02582*, *N-BLR* and *ZEB1-AS1* [23,24,25,26], were excluded from further analysis due to the longer amplification product of pre-designed TaqMan probes.

### 2.3. Reverse Transcription and Quantitative PCR (qPCR)

Associated lncRNAs *LUCAT1*, *MALAT1*, *lncTCF7*, *XIST*, *ZFAS1* and mRNAs for *DCN* (Table 1) were analysed relative to the geometric mean of RGs, *IPO8* and *B2M*. lncRNAs and mRNAs were reverse transcribed using an OneTaq RT-PCR Kit (New England Biolabs, Ipswich, MA, USA) using random primers according to the manufacturer’s instructions. Reverse transcription (RT) reactions were started with 3.0 µL (60 ng) of total RNA and 1.0 µL of Random Primer Mix incubated at 70 °C for 5 min. The 10 μL RT master mix included 5.0 μL of M-MuLV Reaction Mix, 1.0 μL of M-MuLV reverse transcriptase and 4.0 μL of reaction mix after random priming. The reaction conditions were 25 °C for 5 min, 42 °C for 60 min and 80 °C for 4 min.

After cDNA synthesis, a preamplification reaction was performed using the TaqMan PreAmp mastermix (Applied Biosystems; Thermo Fisher Scientific, Inc.) following the manufacturer’s protocol. The TaqMan Gene Expression Assays, listed in Table 1, were pooled, followed by dilution to 0.2× using Tris-EDTA buffer solution, pH 8.0 (Sigma-Aldrich; Merck KGaA). The thermocycling conditions were as follows: 10 min at 95 °C and 10 cycles of 15 sec at 95 °C and 4 min at 60 °C.

For miRNAs, reverse transcription looped primers for specific RT of miRNAs and a MicroRNA TaqMan RT kit (Applied Biosystems, Foster City, CA, USA) were utilised following the manufacturer’s protocol. RNU6B and miR-1247b were used as reference genes (RGs). *miR-200c* was tested relative to the geometric mean of expression of *RNU6B* and *miR-1247b* (Table 1). Briefly, a 10 μL RT reaction master mix was performed with 10 ng of total RNA sample, 1.0 μL of MultiScribe Reverse Transcriptase (50 U/μL), 1.0 μL of Reverse Transcription Buffer (10×), 0.1 μL of dNTP (100 mM), 0.19 μL RNAase inhibitor (20 U/μL), and 2.0 μL of RT primer (5×). The reaction conditions were 16 °C for 30 min, 42 °C for 30 min, 85 °C for 5 min.

qPCR reactions were performed using TaqMan technology with FastStart Essential DNA Probe Master (Roche Diagnostics, Basel, Switzerland). Each qPCR reaction contained appropriately diluted cDNA, 2× FastStart Essential DNA Probe Master (Roche Diagnostics) and 20× TaqMan gene expression assay, listed in Table 1. All qPCR reactions were conducted on a Rotor-Gene Q system (Qiagen GmbH), and each sample was run twice. The thermocycling conditions for gene expression were 10 min at 95 °C and 40 cycles of 15 s at 95 °C and 1 min at 60 °C. For miRNAs, the thermocycling conditions were 95 °C for 10 min, 40 cycles for 15 s at 95 °C and for 60 s at 60 °C. The signal was collected at the endpoint of every cycle.

To calculate the efficiency of qPCR reactions, pools of RNA samples of each group were created. The RNA pools were reverse transcribed, and in the cases of lncRNAs and mRNAs, the pools were preamplified as described above and diluted in 4 steps, ranging from 5- to 625-fold dilution; qPCR reactions were run in triplicate as described above.

### 2.4. Statistical Analysis

To calculate relative gene expression, Cq values were corrected according to Latham et al. [31]. To obtain ∆Cq, the Cq of the gene/lncRNA/miRNA of interest was deducted from the geometric mean of Cq values of the reference genes. In CRC, the mRNAs’, miRNAs’ and lncRNAs’ expression differences were compared between both the central part and invasive front of CRC, the central part of CRC and lymph node metastasis and the central part of CRC and liver metastasis using ΔCq and the Wilcoxon rank test. For all correlations/associations, Spearman rank-order correlation was used. All statistical analyses were performed using SPSS analytical software v24 (IBM Corp., Armonk, NY, USA), with a cut-off of *p* ≤ 0.05.

## 3. Results

### 3.1. Patients and Tissue Samples

In total, we analysed 63 tissue samples from 19 patients with CRC with lymph node and/or liver metastases. There were 13 males and 6 women, with a mean age of 70 ± 14. The invasive front and the central part of CRC were available in all cases, whereas lymph node metastases were available in 15 and liver metastases in 10 cases. Detailed information is provided in Table 2.

### 3.2. Expression of Decorin in the Central Part and the Invasive Front of Primary Colorectal Carcinoma in Comparison to Lymph Node and Liver Metastases

The expression of *DCN* was present in all analysed groups; the results are presented in Figure 1. Gene expression of *DCN* was downregulated in the central part when compared to the invasive front of primary CRC, in lymph nodes or in liver metastases. The downregulation was observed in the central part and the invasive front of CRC compared to lymph node and liver metastasis. None of the comparisons yielded statistically significant results.

### 3.3. Expression of miR-200c in the Central Part and Invasive Front of Primary Colorectal Carcinoma in Comparison to Lymph Node and Liver Metastases

The expression of *miR-200c* was present in all analysed groups and is presented in Figure 2. The expression of *miR-200c* was upregulated, but not significant, in the central part of CRC in comparison to the invasive front. The expression of *miR-200c* was significantly downregulated in the central part of CRC as well as the invasive front when compared either to lymph node or liver metastases (*p* = 0.001 and *p* = 0.008 in the central part and *p* = 0.001 and *p* = 0.008 at the invasive front).

### 3.4. Expression of lncRNAs in the Central Part and the Invasive Front of Primary Colorectal Carcinoma in Comparison to Lymph Node and Liver Metastases

When comparing the central part of CRC to the invasive front, *LUCAT1*, *XIST* and *ZFAS1* were non-significantly upregulated in all groups, while *MALAT1* and *lncTCF7* were non-significantly downregulated.

When comparing lymph node metastases to both the central part and invasive front of CRC, the difference in expression of *MALAT1* was significant in both cases (*p* = 0.041 and *p* = 0.013, respectively) and of *lncTCF7* only when compared to the invasive front (*p* = 0.050). The expression of *LUCAT1* was upregulated in the central part of CRC when compared to the expression in lymph node metastasis. *LUCAT1*, *XIST* and *ZFAS1* did not show any statistically significant change in expression when comparing the central part of CRC or the invasive front to the lymph node metastasis.

Additionally, all investigated lncRNAs were downregulated in the central part and the invasive front of CRC when compared to liver metastasis. The difference in expression of the investigated lncRNAs in the invasive front of CRC was not significant when compared to the expression in liver metastases. Nonetheless, significant differences were observed in the expression of *MALAT1*, *lncTCF7* and *ZFAS1* in the central part of CRC when compared to the expression in liver metastasis (*p* = 0.011, *p* = 0.013 and *p* = 0.007, respectively). The expression of the selected lncRNAs is presented in Figure 3.

### 3.5. Correlations between miR-200c, Target Gene Decorin and the Investigated lncRNAs

*XIST* did not correlate significantly with any of the other investigated lncRNAs, *miR-200c* or *DCN*. *miR-200c* had several statistically significant correlations: a weak positive correlation with *DCN*, *lncTCF7* and *ZFAS1* and a moderate positive correlation with *MALAT1*. Additionally, *DCN* correlated significantly and positively with several of the investigated lncRNAs; a weak positive correlation was observed with *lncTCF7* and *ZFAS1*, while a moderate positive correlation was observed with *MALAT1*. Furthermore, several of the investigated lncRNAs correlated significantly and positively with each other, the strongest being between *MALAT1* and *ZFAS1*. Other comparisons and complete data on correlation coefficients and corresponding *p*-values are summarised in Table 3.

## 4. Discussion

The purpose of this study was to evaluate the expression of *DCN* in the central part and the invasive front of primary CRC in comparison to lymph node and liver metastases. Our results showed that the expression of *DCN* was downregulated in the central part and the invasive front of CRC compared to lymph node and liver metastases. Further, we examined whether the expression of *miR-200c* was negatively correlated to the expression of *DCN* and validated the expression of some lncRNAs as potential sponges for *miR-200c*. The Spearman correlation coefficient showed a positive, statistically significant correlation between the expression of *DCN* and *miR-200c* as well as between lncRNAs *MALAT1*, *lncTCF7* and *ZFAS1* and *miR-200c*. These three lncRNAs were also the only ones that were significantly downregulated in the central part or the invasive front of CRC when compared to metastases.

Previous studies of *DCN* expression in CRC were mostly based on a comparison between the expression in tumorous and healthy mucosa [3,4,32]. Our study, however, compared the expression between the central part and the invasive front of CRC to lymph node and liver metastases. The expression of *DCN* was slightly downregulated in both the central part and invasive front of CRC in comparison to lymph node and liver metastases, but the difference in expression was not significant. There are limited data about the heterogeneity of expression of either protein or the mRNA of *DCN* in different parts of CRC. In our previous study, we compared the expression of *DCN* in the central part to the expression of *DCN* in the invasive front, and this was further confirmed in the present study [10]. Another group analysed the expression pattern of protein DCN in healthy mucosa, primary tumour and liver metastases of CRC [29]. Immunohistochemistry conducted by Reszegi et al. showed that the expression of DCN was downregulated in the stroma of the primary tumour compared to the stroma of healthy mucosa. In the liver metastases, the expression of the protein was the same when compared to the adjacent liver tissue but significantly lower than in the normal colon and primary CRC [29]. Based on these and our observations, we speculated that the decreased expression of protein DCN and slight upregulation of mRNA *DCN* in liver metastases compared to the primary tumour may reflect the regulation at the post-transcription level by ncRNAs.

Regarding the gradient of expression of *miR-200c* in CRC, we did not observe any changes in expression; however, some studies reported upregulation of *miR-200c* in the central part in comparison to the invasive front of CRC [33,34]. In concordance with our results of *miR-200c* expression, Hur et al. [35] also found downregulation of *miR-200c* in primary CRC in comparison to liver metastases. In contrast, another study showed *miR-200c* upregulation in primary CRC compared to liver metastases [36]. However, to the best of our knowledge, this was the first report of *miR-200c* expression in lymph node metastases of CRC in comparison to primary CRC besides our previous study that focused on the epithelial-mesenchymal transition [37]. Furthermore, our results suggest that in contrast to a previous bioinformatics analysis [19], *miR-200c* has a positive influence on *DCN* expression in CRC metastases. We can only speculate that the observed positive correlation between the expression of *miR-200c* and *DCN* suggests that *DCN* is regulated by *miR-200c* in an indirect manner (through another factor or binding of *miR-200c* on promotor region of *DCN*) rather than a direct manner (binding of *miR-200c* to the 3′-UTR of *DCN*).

Similarly, the expression of the three of five investigated lncRNAs positively correlated to the expression of *miR-200c;* therefore, sponging of *miR-200c* by selected miRNAs is probably not the case. Speculatively, all these ncRNAs might be, for instance, under the regulation by the same transcription factor. However, all of the five investigated lncRNAs had already been reported to be upregulated in CRC in comparison to healthy mucosa. Moreover, the expression of all lncRNAs had been positively correlated to the development of metastases [38,39,40,41,42,43,44,45,46,47]. According to the available publications, there is no data regarding a comparison of the expression of investigated lncRNAs in the central part of CRC to that in the invasive front, lymph node and liver metastases of CRC.

Although *LUCAT1* was downregulated in our samples of primary CRC in comparison to liver metastasis, downregulation did not reach statistical significance. The lncRNAs’ expression profiles of CRC tissue from patients with liver metastases to those without metastases revealed that *LUCAT1* is a liver metastasis-associated lncRNA. It was also shown that knockdown of *LUCAT1* could significantly inhibit cells’ invasion, suggesting that it might play an important role in liver metastasis by promoting cells’ invasion [38]. Another study confirmed that *LUCAT1* promoted metastases through the stimulating migration and invasion of CRC cell lines [39].

We observed a significantly downregulated expression of *MALAT1* in the central part of CRC in comparison to lymph node and liver metastases. Similar results were reported by Liu et al., who found the expression of *MALAT1* to correlate to liver metastasis status and to be significantly downregulated in CRC tissue compared to excised liver metastases. They also showed that *MALAT1* was significantly upregulated in primary CRC patients who developed liver metastasis within 5 years of initial diagnosis, compared to the primary CRC of patients with no metastasis. A positive signature comprising of high *MALAT1* also correlated with the progression to high-grade CRC [40]. Our study further supports the observation of a significantly lower expression of lncRNAs in primary CRC in comparison to metastases.

LncRNA *lncTCF7* was found to be downregulated in the central part and the invasive front of CRC when compared to lymph node and liver metastases, respectively. *lncTCF7* is believed to have an essential role in maintaining cancer stem cell self-renewal. Its expression has been shown to correlate with lymph node metastasis and stage, promote invasion and migration of tumour cells [41]. Another study supported this observation and additionally showed its significant association with depth of invasion. *lncTCF7* also promoted proliferation. As such, it might predict progression, facilitate tumour growth and promote the formation of metastases in CRC [42].

However, in our study, we were not able to confirm any statistically significant change in expression for *XIST*, but we observed an insignificant upregulation in primary CRC in comparison to lymph node metastases and no change in the expression in primary CRC in comparison to liver metastases. However, it has been reported that *XIST* expedited and promoted the growth of metastases in CRC [43,44].

Finally, we observed a downregulation of lncRNA *ZFAS1* in primary CRC compared to lymph node and liver metastases. The expression of *ZFAS1* was shown to be lower in primary CRC than in metastasis, and it was shown to be positively correlated with lymph node invasion and pTNM stage [45]. *ZFAS1* was also associated with an aggressive CRC phenotype. Its knockdown inhibited cell proliferation and invasion in vitro and metastases in vivo [46]. Furthermore, a high expression of *ZFAS1* was observed in advanced stages of CRC; its silencing reduced the cells’ migration and invasion abilities, supporting the function of *ZFAS1* in the development of metastases of CRC [47].

There are two main limitations of our study. The first is the small number of patients with an unequal female-to-male ratio. However, this was the consequence of collecting the samples from the same patients with CRC from primary tumour, lymph node and/or liver metastases. The second one is the lack of functional validation of regulation of *DCN* by *miR-200c*. However, our results suggest that this could be a rather complex investigation since we observed the positive rather than expected negative correlation between *miR-200c* and *DCN*.

## 5. Conclusions

In conclusion, there are three main findings from our study. First, we found an insignificant downregulation of the *DCN* gene in primary CRC in comparison to metastases, which, regarding the reported upregulated protein expression in primary CRC in comparison to metastases, suggests that there is a post-transcriptional level of the regulation of *DCN*. Second, we observed a significant downregulation of the investigated ncRNAs in primary CRC in comparison to metastases, suggesting their important role in the development of metastases. Finally, the observed positive regulation between the expression of all these RNAs suggests that *DCN* and *miR-200c* might be regulated in an indirect rather than a direct manner. However, our study provides further evidence of an important role not only of *DCN* and *miR-200c* but also of lncRNAs *MALAT1*, *lncTCF7* and *ZFAS1* in the development of metastases in CRC.

## Figures and Tables

**Figure 1 biomedicines-10-00142-f001:**
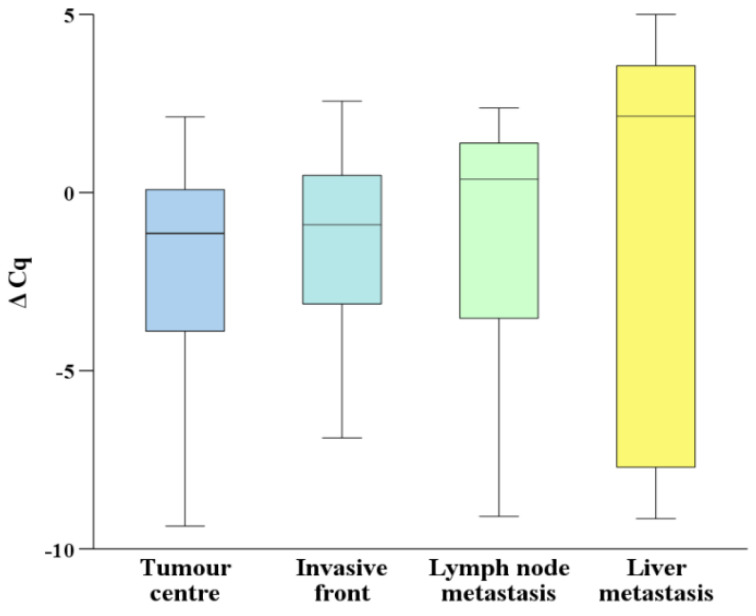
Expression of decorin in the central part and the invasive front of primary colorectal carcinoma in lymph node and liver metastases. Legend: ΔCq, delta Cq.

**Figure 2 biomedicines-10-00142-f002:**
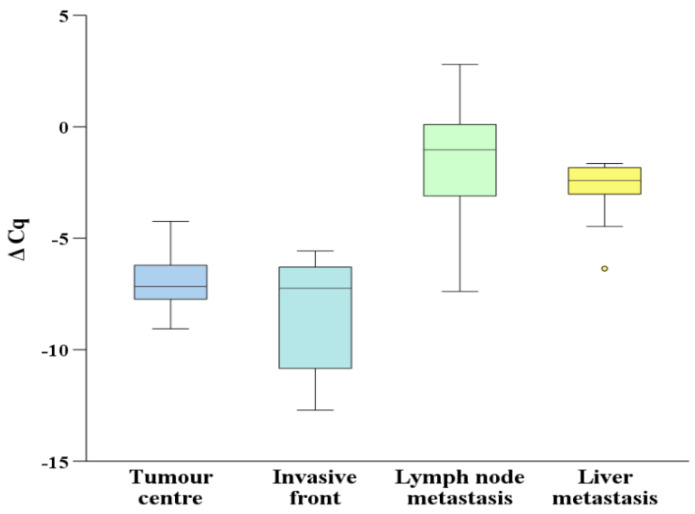
Expression of *miR-200c* in the central part and the invasive front of primary colorectal carcinoma in lymph node and liver metastases. Legend: ΔCq, delta Cq; °, high potential outlier as defined by SPSS.

**Figure 3 biomedicines-10-00142-f003:**
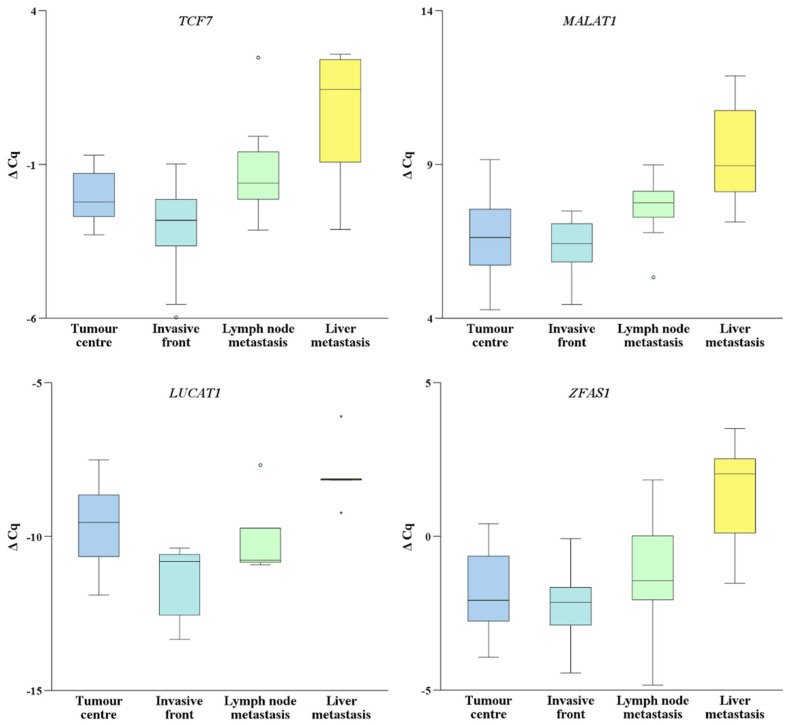
Expression of lncRNAs in the central part and the invasive front of primary colorectal carcinoma in lymph node and liver metastases. Legend: ΔCq, delta Cq; °, high potential outlier as defined by SPSS; *, high extreme values as defined by SPSS.

**Table 1 biomedicines-10-00142-t001:** List of used TaqMan gene expression assay.

Gene or lncRNA	Assay ID
*B2M*	Hs99999907_m1
*IPO8*	Hs00183533_m1
*DCN*	Hs00266491_m1
*LUCAT1*	Hs04978593_m1
*MALAT1*	Hs0191077_s1
*lncTCF7*	Hs01556515_m1
*ZFAS1*	Hs01379985_m1
*XIST*	Hs01077162_m1
*RNU6B*	ID 001093
*miR-1274b*	ID 002884
*miR-200c*	ID 002300

**Table 2 biomedicines-10-00142-t002:** Patients’ characteristics and pTNM status.

Group	Age (Mean ± SD)	Male:Female	pTNM	Tissue Samples
N+ M0	74.2 ± 13.4	7:2	pT3N1 (*n* = 6) pT4N2M0 (*n* = 1) pT4N2M1 (*n* = 2)	Invasive front (*n* = 9) Central part (*n* = 9) Lymph node metastasis (*n* = 9)
N0 M+	70.5 ± 4.9	2:2	pT1 (*n* = 1) pT3N0 (*n* = 1) pT3N0M1 (*n* = 1) pT4aN1a (*n* = 1)	Invasive front (*n* = 4) Central part (*n* = 4) Liver metastasis (*n* = 4)
N+ M+	63.8 ± 15.5	4:2	pT3N1 (*n* = 1) pT3N2M1 (*n* = 1) pT4N1M1 (*n* = 4)	Invasive front (*n* = 6) Central part (*n* = 6) Lymph node metastasis (*n* = 6) Liver metastasis (*n* = 6)

**Table 3 biomedicines-10-00142-t003:** Significant Spearman correlation coefficients and corresponding *p*-values for investigated comparisons between the *miR-200c*, decorin and analysed lncRNAs.

Correlations and *p*-Values	*DCN*	*miR-200c*	*LUCAT1*	*MALAT1*	*XIST*	*lncTCF7*	*ZFAS1*
*DCN*	1	0.296 (0.027)	/	0.452 (<0.001)	/	0.278 (0.034)	0.357 (0.006)
*miR-200c*	0.296 (0.027)	1	/	0.425 (0.001)	/	0.313 (0.018)	0.375 (0.004)
*LUCAT1*	/	/	1	0.633 (<0.001)	/	0.613 (0.001)	0.516 (0.006)
*MALAT1*	0.452 (<0.001)	0.425 (0.001)	0.633 (<0.001)	1	/	0.741 (<0.001)	0.746 (<0.001)
*XIST*	/	/	/	/	1	/	/
*lncTCF7*	0.278 (0.034)	0.313 (0.018)	0.613 (0.001)	0.741 (<0.001)	/	1	0.641 (<0.001)
*ZFAS1*	0.357 (0.006)	0.375 (0.004)	0.516 (0.006)	0.746 (<0.001)	/	0.641 (<0.001)	1

## Data Availability

The data presented in this study are available on request from the corresponding author.

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
