# Peer review of "Differential Expression of Decorin in Metastasising Colorectal Carcinoma Is Regulated by miR-200c and Long Non-Coding RNAs"

_biomedicines, 2022, doi:10.3390/biomedicines10010142_

Round 1

Reviewer 1 Report

Although it is hard to collect metastatic CRC samples, the patient & sample number here is rather low (10 patients and 63 tissue samples). The analogy male/female is also not 50%.

There is no statistical significance to support that "the upregulation was on an upward trend from the central part, invasive front, lymph node metastasis to liver metastasis."

Although the authors state "The expression of miR-200c was statistically significantly downregulated in the central part of CRC as well as invasive front when compared either to lymph node or liver metastases", in Fig. 2 miR-200c seems to be upregulated in the central part and invasive front.

Please avoid using phrases like "statistically non-significantly downregulated". "Not significantly downregulated" would be fine.

There is no point in presenting the non-significant expression of lncRNAs in Fig. 3. 

What do the positive correlations between the expression of miR-200c, decorin, and the investigated lncRNAs in paragraph "3.4" tell us? Please elaborate.

There is no in-vitro validation in the proposed miR-200c/DCN/lncRNAs axis.

Author Response

Although it is hard to collect metastatic CRC samples, the patient & sample number here is rather low (10 patients and 63 tissue samples). The analogy male/female is also not 50%.

We agree that the number of samples is low; however, there were 19 and not 10 patients. For such type of the study (collecting primary tumor, lymph node and liver metastasis of the same patients) it is also difficult to collect the sample where female : male ratio is 1 : 1. We added this observation as limitation of the study.

There is no statistical significance to support that "the upregulation was on an upward trend from the central part, invasive front, lymph node metastasis to liver metastasis."

The sentence was corrected according to results.

Although the authors state "The expression of miR-200c was statistically significantly downregulated in the central part of CRC as well as invasive front when compared either to lymph node or liver metastases", in Fig. 2 miR-200c seems to be upregulated in the central part and invasive front.

Yes, we agree. However, this discrepancy is due to philosophy of calculating ΔCq, where geomean of RGs is subtracted from GOI. However, to avoid confusing presentation of the data, we subtracted the GOI from geomean of RGs. We corrected all the figures.

Please avoid using phrases like "statistically non-significantly downregulated". "Not significantly downregulated" would be fine.

We corrected this through the manuscript.

There is no point in presenting the non-significant expression of lncRNAs in Fig. 3. 

We removed the lncRNA XIST, since it was the only lncRNA that was not differentially expressed either among different parts of tumour or in correlation to DCN or miR-200c.

What do the positive correlations between the expression of miR-200c, decorin, and the investigated lncRNAs in paragraph "3.4" tell us? Please elaborate.

We can only speculate the positive correlation. As we already suggested in the conclusion of the manuscript, the positive correlation between miR-200c and DCN suggests that DCN is regulated by miR-200c in an indirect manner (through another factor or binding of miR-200c on promotor region of DCN) rather than through a direct manner (binding of miR-200c on 3-UTR of DCN). Similarly, sponging of miR-200c by selected miRNAs is probably not the case, but might be, for instance, under the regulation by the same transcription factor. We added this speculation in the discussion section.

There is no in-vitro validation in the proposed miR-200c/DCN/lncRNAs axis.

Indeed, there is no in vitro validation for miR-200c regulating the DCN; however, this is the first report on qPCR validation in CRC samples, beside reported correlation in TCGA samples. It would be far more complicated to address such an important and highly technologically demanding question, which was raised and committed to us by one of the two reviewers, in the required 10 days timeframe, and to do all the experiments which are necessary  to verify the miRNA (like miR-200c) targets in vitro. Nevertheless, we have some concerns regarding required experiments that are described below. Numerous control experiments needs to be performed, (i) plasmid vectors without 3'-UTR, with 3'-UTR insert in the reverse orientation, and 3'-UTR without miRNA binding site of interest, (ii) co-transfection of control luciferase reporter plasmid for normalization of data for experimental variation in transfection efficiencies (1). Reporter assays can result in misleading assessment of targets since transfection of supra-physiological concentrations of miRNAs may create a situation in which two molecules with complementary surfaces may engage in non-physiological interactions. Expression of the 3'-UTR and the tested miRNA in heterologous context may lead to non-physiological interactions because of the incorrect cofactor environment. However, optimization of miRNA transfection conditions for cell type utilized (e.g. fluorescent labelled miRNAs) should be performed, as well as specific and scrambled double-stranded RNA control molecules should be used. In addition, mRNA turnover rate limits miRNA efficacy (2), and in miRNA experiments strong confounding factors are (i) target gene expression and (ii) competition between endogenous and transfected miRNAs (3). Target regulation is under the influence of temporal and spatial-specific mechanisms, therefore cell type, differentiation state of cell, and whether a cell is under stress all is believed to influence miRNA regulation of target (4).

Several other approaches exist to validate miRNA/mRNA interactions, without the use of reporter assay. An example is pull-down strategy, miRNA and target mRNA co-expression, gain-of-function (miRNA mimics) in cell type known to express the putative target protein (1). We are highly aware that this is only the first step of deeper understanding such of an important function of lncRNA-miRNA-DCN interaction, that can be used as a framework for further verification, performed either by us or other interested, more cell-lines specialized research groups. However, miR-200c being sponged by selected lncRNAs was already shown in cancerogenesis, although not all of them in colorectal cell lines (5-9).

We added this observation as limitation of the study.

References

(1) Kuhn, D.E.; Martin, M.M.; Feldman, D.S.; Terry, A.V. Jr; Nuovo, G.J.; Elton, T.S. Experimental validation of miRNA targets. Methods 2008, 44, 47-54.

(2) Larsson, E.; Sander, C., Marks, D. mRNA turnover rate limits siRNA and microRNA efficacy. Mol Syst Biol 2010, 6, 433.

(3) Saito, T.; Sætrom, P. Target gene expression levels and competition between transfected and endogenous microRNAs are strong confounding factors in microRNA high-throughput experiments. Silence 2012,3, 3.

(4) van Rooij, E. The art of microRNA research. Circ Res 2011, 108, 219-34.

(5) O'Brien, S.J.; Fiechter, C.; Burton, J.; Hallion, J.; Paas, M.; Patel, A.; Patel, A.; Rochet, A.; Scheurlen, K.; Gardner, S.; et al. Long non-coding RNA ZFAS1 is a major regulator of epithelial-mesenchymal transition through miR-200/ZEB1/E-cadherin, vimentin signaling in colon adenocarcinoma. Cell Death Discov 2021, 7, 61.

(6) Meng, L.; Ma, P.; Cai, R.; Guan, Q.; Wang, M.; Jin, B. Long noncoding RNA ZEB1-AS1 promotes the tumorigenesis of glioma cancer cells by modulating the miR-200c/141-ZEB1 axis. Am J Transl Res 2018, 10, 3395-3412.

(7) Zhang, M.; Wang, F.; Xiang, Z.; Huang, T.; Zhou, W.B. LncRNA XIST promotes chemoresistance of breast cancer cells to doxorubicin by sponging miR-200c-3p to upregulate ANLN. Clin Exp Pharmacol Physiol 2020, 47, 1464-1472.

(8) Han, Z.; Shi, L. Long non-coding RNA LUCAT1 modulates methotrexate resistance in osteosarcoma via miR-200c/ABCB1 axis. Biochem Biophys Res Commun 2018, 495, 947-953.

(9) Pa, M.; Naizaer, G.; Seyiti, A.; Kuerbang, G. Long noncoding RNA MALAT1 functions as a sponge of miR-200c in ovarian cancer. Oncol Res 2017.

Reviewer 2 Report

There are correlations between histological invasiveness and DCN content?  Could you explain a greater content of DCN in the peripheral part of the tumor?  Did you find correlations with immunological cellular reaction to the CRC?

Author Response

There are correlations between histological invasiveness and DCN content? Could you explain a greater content of DCN in the peripheral part of the tumor?  Did you find correlations with immunological cellular reaction to the CRC?

There are several studies reporting on immunohistochemical analysis of decorin. The expression of decorin was shown to be present in stromal cells, the expression being the strongest in healthy mucosa and the lowest in carcinomas (1, 2, 3). However, our immunohistochemical analysis also showed that expression of decorin is present in stromal cells, but is higher in carcinoma. According to the available literature and our analysis, there is no expression of decorin in tumour cells and we did not find any differences in immunohistochemical staining between central parts of the tumour and invasive front (4).

Our previous study showed that in contrast to immunohistochemical expression of decorin, the expression of the decorin gene showed higher expression at the invasive front when compared to the central part. Moreover, a detailed analysis of expression in CRC showed that expression of decorin was upregulated at the invasive front of the tumour compared with the central parts in the CRC with lymph node metastasis, whereas in the CRC without lymph node metastasis, expression at the invasive front was similar to the central parts. The difference in expression was not significant (4). To the best of our knowledge there was no study conducted on CRC or other carcinomas studying the heterogeneity of decorin expression in cancer.

We further observed that decorin was significantly downregulated in CRC with lymph node metastasis compared to CRC without lymph node metastasis (4).

Transcription and translation of gene DCN is present in fibroblast, endothelial cells and smooth-muscle cells (5). Decorin is widely accepted as regulator of collagen fibrilogenesis (1, 3), but also contribute to migration, proliferation, apoptosis and differentiation (6). Decorin binds growth factors such as TGF-b, and inhibits tyrosine kinase receptors such as EGFR, IGF in MET (5, 7). Binding complex decorin-TGF-beta reduces effect of TGF-beta and consequently the amount of fibrotic tissue. Decorin is believed to be one of the regulators of the synthesis of the ECM components, inhibitor of the maturation of the collagen I, it contributes to the expression of collagenase and consequently to the proliferation of vasculature (8).

There is limited data in the literature about relationship between extracellular matrix and tumour infiltrating lymphocytes in cancer, particularly in colorectal cancer. Moreover, there are no data on correlation between decorin expression and tumour infiltrating lymphocytes. Due to the limited number of patients involved in our study (19 patients and 63 tissue samples) that was a consequence of the collection of patients with lymph node and/or liver metastases, we do not believe that analysing the correlation between immunological cellular reaction and expression of decorin would provide relevant results.

(1) M. C. Nyman, A. O. Sainio, M. M. Pennanen, R. J. Lund, S. Vuorikoski, J. T. Sundstrom, et al. Decorin in human colon cancer: localization in vivo and effect on cancer cell behavior in vitro. J. Histochem. Cytochem. 2015; 63(9):710-20.

(2) A. Reszegi, Z. Horvath, K. Karaszi, E. Regos, V. Postnikova, P. Tatrai, et al. The protective role of decorin in hepatic metastasis of colorectal carcinoma. Biomolecules. 2020; 10(8):1199.

(3) K. Augoff, J. Rabczynski, R. Tabola, L. Czapla, K. Ratajczak, K. Grabowski, et al. Immunohistochemical study of decorin expression in polyps and carcinomas of the colon. Med. Sci. Monit. 2008; 14(10):CR530-5.

(4) M. Žlajpah, N. Hauptman, E. Boštjančič, N. Zidar. Differential expression of extracellular matrix-related genes DCN, EPHA4, FN1, SPARC, SPON2 and SPP1 in colorectal carcinogenesis. Oncol. Rep. 2019; 42(4):1539-1548.

(5) T. Neill, L. Schaefer, R. V. Iozzo. Decorin: a guardian from the matrix. Am. J. Pathol. 2012; 181(2):380-7.

(6) Z. Liu, Y. Yang, X. Zhang, H. Wang, W. Xu, H. Wang, et al. An oncolytic adenovirus encoding decorin and granulocyte macrophage colony stimulating factor inhibits tumor growth in a colorectal tumor model by targeting pro-tumorigenic signals and via immune activation. Hum. Gene Ther. 2017; 28(8):667-80.

(7) K. Moreth, R. V. Iozzo, L. Schaefer. Small leucine-rich proteoglycans orchestrate receptor crosstalk during inflammation. Cell Cycle. 2012; 11(11):2084-91.

(8) W. Zhang, Y. Ge, Q. Cheng, Q. Zhang, L. Fang, J. N. Zheng, et al. Decorin is a pivotal effector in the extracellular matrix and tumour microenvironment. Oncotarget. 2018; 9(4):5480-91.

Round 2

Reviewer 2 Report

I agree with your revisions.